# Plasma SerpinA5 in conjunction with uterine artery pulsatility index and clinical risk factor for the early prediction of preeclampsia

**Yonggang Zhang**[1]*, **Yipeng Zhang**[1], **Limin Zhao**[1], **Junzhu Shi**[1], **Hongling Yang**[2]

**1** Department of Clinical Laboratory, Shenzhen Longhua District Central Hospital, Guangdong Medical University, Shenzhen, Guangdong, China, **2** Department of Clinical Laboratory, Guangzhou Women and Children's Medical Centre, Guangzhou Medical University, Guangzhou, Guangdong, China

* 1093629149@qq.com

**Data Availability Statement:** All relevant data are within the paper and its Supporting Information files.

## Abstract

### Object

This study aimed to combine plasma protein SerpinA5 with uterine artery doppler ultrasound and clinical risk factor during the first trimester for prediction of preeclampsia.

### Methods and materials

This study was a nested cohort study and was divided into the screening set and developing set. The plasma was collected during the first trimester ($11^{+0}$–$13^{+6}$ weeks), at the same time, UtA-PI was detected and recorded with four-dimensional color Doppler ultrasound. These pregnancies were followed up until after delivery. The plasma proteins were examined using ultra-performance liquid chromatography–mass spectrometry (UPLC-MS) and enzyme linked immunosorbent assay (ELISA). Placental samples preserved after delivery were analysed by immunohistochemistry. Clinical risk factors were obtained from medical records or antenatal questionnaires. Upregulation or downregulation of SerpinA5 expression in TEV-1 cells was performed to investigate the role of SerpinA5 in trophoblasts invasion.

### Results

We demonstrated that SerpinA5 levels were greater not only in preeclampsia placental tissue but also in plasma (both $p < 0.05$), and we found that SerpinA5 may interfere with trophoblastic cell invasion by inhibiting MSP. SerpinA5 may be a potential predictor of preeclampsia. What is more, the sensitivity and specificity of predictive power were strengthened when plasma SerpinA5 was combined with UtA-PI and pre-pregnancy BMI & family history of PE for prediction of preeclampsia.

### Conclusion

These findings showed that placenta-derived plasma SerpinA5 may be a novel biomarker for preeclampsia, which together with uterine artery Doppler ultrasound and clinical risk factor can more effectively predict preeclampsia.

**Funding:** This study was supported by the National Natural Science Foundation of China (81801474), the National Natural Science Foundation of China (81871716), and the Science and Technology Fund of Shenzhen (JCYJ20180306172502097). There are no conflicts of interest for any of the funding sources.

**Competing interests:** The authors report no conflicts of interest.

## Introduction

Preeclampsia (PE) is a common and important obstetric disease. PE occurs in 5–7% of all pregnancies and is a leading cause of maternal mortality. It is reported that apparently 16% of maternal deaths in developed countries are associated with PE and up to 25% in developing countries [1]. Compared to the healthy population, PE patients and their infants are likely to have a four-fold higher risk of elevated blood pressure and diabetes within 10–20 years [2]. Screening for PE before the onset is important for identifying at-risk patients who might benefit from close follow-up and potential treatment to prevent adverse pregnancy outcomes [3, 4]. Given that multi faceted joint forecasting may play a role in the management of preeclampsia [2, 5], the union of placental-derived plasma protein, color Doppler ultrasound examination and epidemiological investigation may be a research strategy for preeclampsia.

We screened 20 proteins in plasma that were different between women with preeclampsia and normal pregnant women and may contribute to early monitoring and prevention of preeclampsia. Through bioinformatics prediction (GO analysis), we found that SerpinA5 (Plasma Serine Protease Inhibitor, Clade A, Member 5), which is a protein C inhibitor, was the core protein. In the current study, we studied the relationship between SerpinA5 and preeclampsia and whether SerpinA5 can be used as a potential biomarker for preeclampsia, and a multi-element prediction model for preeclampsia was established.

## Materials and methods

This study was a nested cohort study. All women who received a prenatal examination and subsequently delivered between July 2015 and June 2018 were enlisted at Shenzhen Longhua District Central Hospital (SZLHCH), China. The hospital ethics committee of SZLHCH approved all aspects of this study. Written informed consent was obtained from all subjects and they were anonymous and numbered. Preeclampsia (PE) was defined as new-onset hypertension after 20 weeks of gestation and proteinuria or non-proteinuria but with other serious features according to American College of Obstetricians and Gynecologists (ACOG) 2013. Blood pressure was documented on 2 separate occasions that were at least 6 hours apart and resolved at the 6th week postpartum. For the control group, participants exhibited normal blood pressure without an excess of protein in the urine, pregnancy complications or other foetal malformations. Among the subjects who subsequently developed PE, we excluded women who had twin and multiple pregnancies, stillbirth, gestational diabetes mellitus, uterine leiomyoma or hyperthyroidism.

A total of 5000 pregnant women were recruited, of whom 60 gravidas developed PE after 20 weeks of gestation and satisfied all of the inclusion and exclusion criteria for the PE group. We randomly divided the PE cases into the screening set (12 PE gravidas) and developing set (48 PE gravidas). In the two sets, all the gravidas who subsequently developed PE were matched at a 1:1 ratio for age, gestational week and sample date to controls who had a pregnancy without complications. The details of the plasma study design are shown in **Fig 1**. Maternal whole blood samples were obtained from all participants during the first trimester ($11^{+0}$–$13^{+6}$ weeks) between July 2015 and June 2018. Fresh peripheral venous blood samples (2 ml) were collected with EDTA and then centrifuged (3000 rpm for 10 min at room temperature). Subsequently, the plasma was immediately separated and stored at -80˚C until use. Placental samples were taken from the maternal-foetal interface near the maternal side and stored in a liquid nitrogen tank after the placentas were removed by Caesarean section or natural delivery (**Table 1**). Plasma (collected during the first trimester ($11^{+0}$–$13^{+6}$ weeks) were examined using ultra-performance liquid chromatography–mass spectrometry (UPLC-MS) and enzyme linked immunosorbent assay (ELISA). Placental tissue was analysed using immunohistochemical methods.

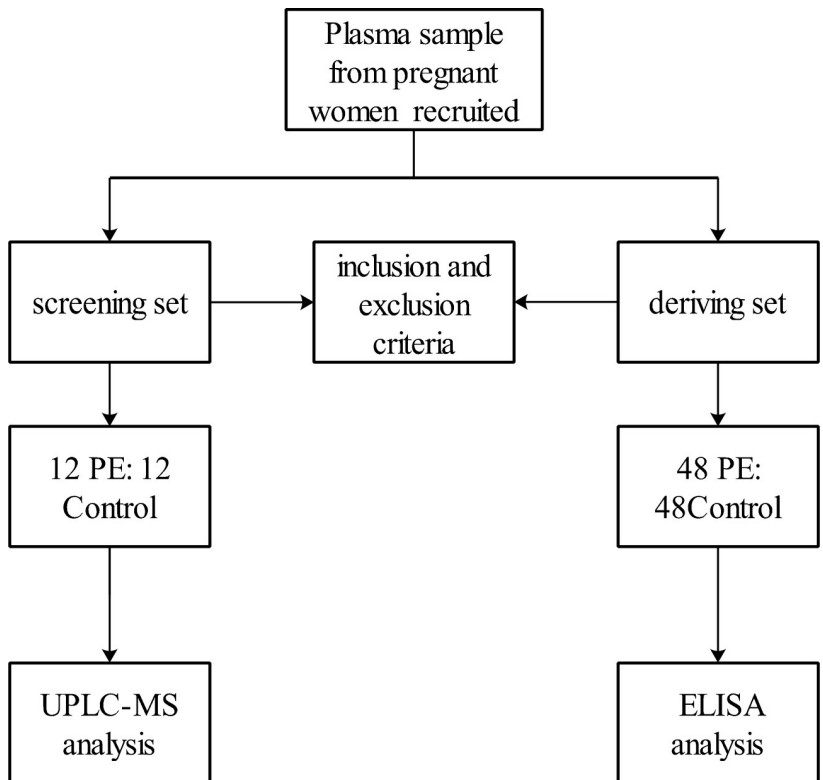

**Fig 1. Flow diagram of the participants in the plasma study.**

All experiments were carried out in accordance with the approved guidelines and regulations, in line with the tenets of the Declaration of Helsinki.

## Analysis of plasma

In the screening set, the proteome of plasma was assessed using ultra-performance liquid chromatography–mass spectrometry (UPLC-MS). One hundred microlitres of plasma samples was

**Table 1. Clinical characteristic of patients and control group.**

| Characteristic | PE (n = 48) | Control (n = 48) | P-value |
|---|---|---|---|
| Maternal Age (year), n = 60:60 | 32.33 ± 2.65 | 31.66 ± 3.51 | 0.26 |
| Gestational age (week) | 35.61 ± 3.43 | 36.52 ± 2.37 | 0.082 |
| Systolic (mmHg) | 157.19 ± 13.69 | 117.11 ± 7.46 | < 0.001 |
| Diastolic (mmHg) | 103.48 ± 6.85 | 77.22 ± 6.47 | < 0.001 |
| Proteinuria (> 0.3g/24 h n, %) | 100 (100%) | 0 (0%) | < 0.05 |
| Smoking (n, %) | 0 (0%) | 0 (0%) | > 0.05 |
| New-born weight (g) | 2511.08± 754.19 | 3524.1 ± 635.09 | < 0.001 |
| PE family history (n, %) | 5(10.42%) | 0(0%) | < 0.05 |
| Pre-pregnancy BMI (kg/m$^2$) | 26.19 ± 1.54 | 24.42 ± 1.47 | 0.013 |
| The average UtA-PI | 2.16 ±0.67 | 1.65±0.39, | 0.027 |
| Levels of SerpinA5 (ng/ml) | 0.29 ±0.27 | 0.07±0.03 | < 0.001 |
| Caesarean section (n, %) | 30(62.50%) | 15(31.25%) | < 0.001 |
| Fetal Growth Restriction (n, %) | 0 (0%) | 0 (0%) | / |

PE, preeclampsia; BMI, body mass index

dissolved in PBS, and the samples were then extracted with 1 mL of ethyl acetate (twice). The ethyl acetate layer was separated and evaporated under reduced pressure. The obtained sample was then reconstituted in 1 mL of methanol and transferred to a UPLC vial, and aliquots of 10 µL of the sample were injected into the UPLC–MS system. The UPLC–MS system operation was carried out according to the guidelines of the machine instructions.

In the developing set, the level of SerpinA5 was evaluated in plasma by commercially available ELISA kits (R&D Systems). The assay was carried out adhering to the manufacturer's instructions.

## Immunohistochemical trial

SerpinA5 was detected using a rabbit anti-human SerpinA5 antibody (catalogue no: ab228632, Abcam, Cambridge County, UK) and an anti-rabbit cell and tissue staining kit, REAL™ EnVision+/HRP RABBIT (catalogue no: K5007; Dako Denmark A/S, Denmark), as previously described with modifications [6]. Antigen retrieval was performed using EDTA buffer and microwave heating. Staining for SerpinA5 was based on the formation of a horseradish peroxidase (HRP) and anti-rabbit antibody (secondary antibody) complex bound to a rabbit anti-human antibody (primary antibody) targeting SerpinA5. Visualization was based on enzymatic conversion of a chromogenic substrate, 3,3'-diaminobenzidine (DAB), into a brown precipitate by horseradish peroxidase at the sites of SerpinA5 antigen localization. A separate antibody preparation with a SerpinA5 rabbit anti-human antibody (1:150 dilution) was made for each sample. Digital photomicrographs (ten randomly selected fields per section and three sections each specimen) were obtained for use in grading the identified normal and PE sections. The stained sections were evaluated for SerpinA5 expression by 2 investigators who were blinded to the histologic diagnoses using Image-Pro Plus 6.0 software (Media Cybernetics, Inc. Rockville, MD, USA). The average optical density and the corresponding brown-yellow-positive area were provided with each photomicrograph. Finally, each group was represented by the average optical density (IOD [mean]).

## Invasive trial

This invasive trial included three groups: the control group, the SerpinA5- overexpressed(SerpinA5-OE) group, the SerpinA5-knockdown (SerpinA5-KD) group. Matrigel invasion assays were performed as previously described with slight modifications [7]. Briefly, liquid Matrigel (BD, USA) and serum-free DMEM-F12 medium were mixed at a ratio of 1:9. We plated 200 µL of transfected TEV-1 cells (1x106 cells/ml) onto 8-µm pore size ECMatrix gel-coated cell culture inserts in 24-well plates (Chemicon, Temecula, CA); the wells contained 500 µl of culture medium containing 20% serum. After incubation at 37˚C in 5% CO2 for 48 h, the non-invading cells and the ECMatrix gel from the upper surface of the inserts were removed using a cotton swab. The cells on the lower surface of the membrane at the bottom of the well were fixed using methanol and stained with 0.4% crystal violet for 10 min. The cells were then washed quickly three times with saline. After the membranes were dried, the number of cells was counted in eight random microscope fields (at 100X magnification). Each assay was repeated three times. The results are expressed as the infiltrate index (a percentage of the control).

## Color Doppler ultrasonic analysis

UtA-PI was measured using transabdominal color Doppler ultrasonic diagnostic system (GE Medical Systems, Wuxi, China). When a pregnant woman was supine with the bladder was slightly filled. First, the cervix was located. Next, the probe was moved to the horizontal plane

of the interface between the uterine body and cervix and then the ascending branch of uterine artery was shown. The position of first recurve up of the cervix isthmus was sampling points of gate pulse Doppler. The UtA-PI was recorded and an average UtA-PI > the 95th percentile for each pregnant age was regarded as abnormal result.

### Clinical epidemiological data collection

Clinical risk factors were obtained from the Medical Records, prenatal and postpartum questionnaires from Mar 2015 to Sep 2019.

### Statistics

Stata23.0 was used. Continuous variables were compared for comparison of differences between healthy pregnant women and PE patients with a two-tailed Student's t-test or Mann–Whitney U test. Categorical variables were compared using chi-square test or Fisher's exact test. Volcano plot analysis shows the differentially expressed proteins between PE and control subjects using fold change and $P < 0.05$. Receiver operating characteristic (ROC) curves, the sensitivity, specificity, and area under the ROC curve (AUC) were calculated by Medcalc v19.6.1 (MedCalc Software bvba, Ostend, Belgium).

## Results

### Clinical characteristics and color Doppler ultrasonic

Clinical characteristic of patients and control group was shown in **Table 1**. There were no significant differences except for obesity (Pre-pregnancy BMI) and PE family history. The average UtA-PI of preeclampsia was significantly higher than that control during the first trimester (2.16 ±0.67 vs. 1.65±0.39, p = 0.027) (**Table 1**).

### Proteomics and ELISA

Initially, in the screening set, according to analysis of volcano plots (**Fig 2A**), we found that 20 plasma proteins were significantly different between the PE and control groups using UPLC-MS (**Table 2**, n = 12:12). Through bioinformatics prediction (GO analysis), we showed that SerpinA5 was a crucial protein. In the developing set, we demonstrated higher SerpinA5 levels in plasma of preeclampsia than in the control via ELISA (0.293 ±0.266 ng/ml vs. 0.074 ±0.033 ng/ml, P<0.001) (**Fig 2B**, n = 48:48).

### SerpinA5 immunohistochemistry

We also demonstrated that the staining of SerpinA5 (average optical density (IOD [mean]) in preeclampsia placental tissue was greater than that in the control (**Fig 3**, P<0.05).

### In vitro invasion analysis

Furthermore, mechanistically, we found that over expressed SERPINA5 could inhibit the invasion of trophoblasts; knockdown SERPINA5 could enhance invasion of trophoblasts (**Fig 4**). SerpinA5 may be able to negatively regulate the macrophage stimulating protein (MSP) and MSP could be the target of SERPINA5 (**Fig 5**). Which showed that SerpinA5 may interfere with trophoblastic cell invasion by inhibiting MSP.

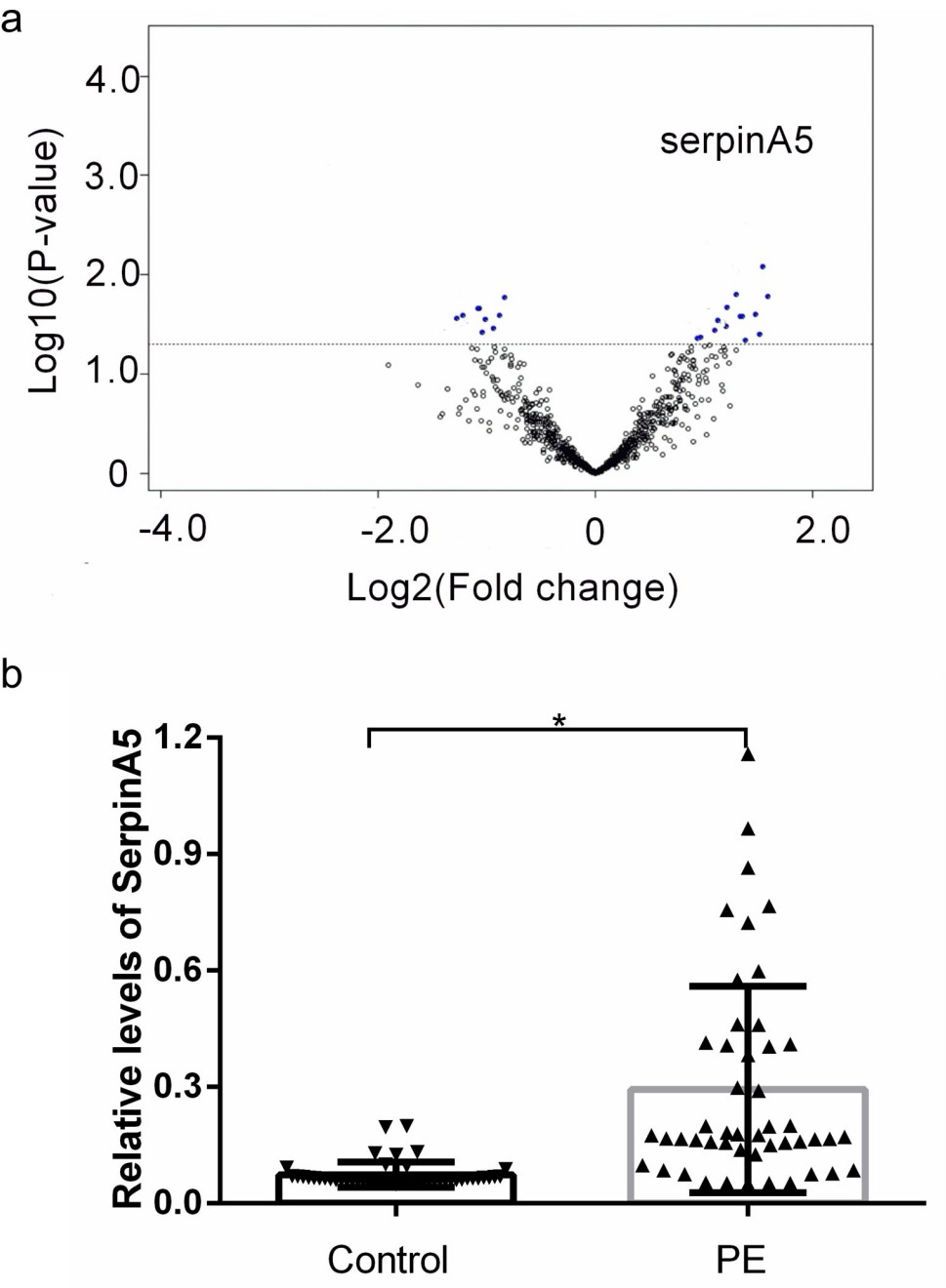

**Fig 2. a. The results of Proteomics in the screening set.** Twenty plasma protein (12 upregulated [including SerpinA5], 8 downregulated) levels were different between the PE group (n = 12) and the control group (n = 12) in the screening set using volcano plots (fold change and p-value). **b. The results of ELISA in the developing set.** The level of SerpinA5 in plasma was significantly higher in the patients with preeclampsia (0.293± 0.266, ng/ml, n = 48) compared to that of the controls (0.074± 0.033 ng/mL, n = 48) before the 20th week of gestation (p<0.001, Student's t-test).

## Prediction analysis

Finally, based on the results of protein screening and developing experiments together with demographic characteristics and Doppler ultrasonic, we performed an analysis to predict

**Table 2. Twenty proteins were significantly different in plasma exosomes between the PE and control groups (12 upregulated; 8 downregulated).**

| Uniprot ID | Protein name | Fold change | P value |
|---|---|---|---|
| P17948 | Vascular endothelial growth factor receptor 1 | 3 | 0.013 |
| P02679 | Fibrinogen gamma chain | 2.98 | 0.0113 |
| Q9UHV8 | Galectin (Placental Protein 13) | 2.75 | 0.0133 |
| P17813 | Endoglin | 2.73 | 0.0121 |
| *P05154* | *Plasma serine protease inhibitor (SerpinA5)* | *2.72* | *0.0148* |
| P01019 | Angiotensinogen | 2.71 | 0.016 |
| P05362 | Intercellular adhesion molecule 1 | 2.68 | 0.0163 |
| P10909 | Clusterin | 2.55 | 0.0151 |
| Q8WXF3 | Relaxin-3 | 2.39 | 0.0169 |
| P05783 | Cytokeratin 18 | 2.29 | 0.0184 |
| P02751 | Fibronectin | 2.19 | 0.0135 |
| Q08380 | Galectin 3 binding protein | 2.04 | 0.0161 |
| Q6IB04 | Placental growth factor protein | 0.50 | 0.014 |
| P18065 | Insulin-like growth factor-binding protein 2 | 0.52 | 0.016 |
| Q13219 | Pappalysin | 0.53 | 0.016 |
| P15692 | Vascular endothelial growth factor A | 0.57 | 0.015 |
| P05155 | Plasma protease C1 inhibitor | 0.58 | 0.011 |
| P27824 | Calnexin | 0.62 | 0.015 |
| P14780 | Matrix metalloproteinase-9 | 0.62 | 0.015 |
| P08253 | Matrix metalloproteinase-2 | 0.63 | 0.015 |

preeclampsia. The area under the receiver operating characteristic (ROC) curve (AUC) of SerpinA5 was 0.881 (95% CI 0.805–0.956). The sensitivity was 89.58 and the specificity was 79.22%, with the 0.070 ng/ml cut-off value (**Fig 6A**). A combination of plasma serpinA5, maternal factors and abnormal UtA-PI (above the 95th percentile) enhanced the predictive value for preeclampsia (United AUC was 0.946 (95% CI 0.905–0.988), **Fig 6B**) and their sensitivity, specificity was shown in **Table 3**.

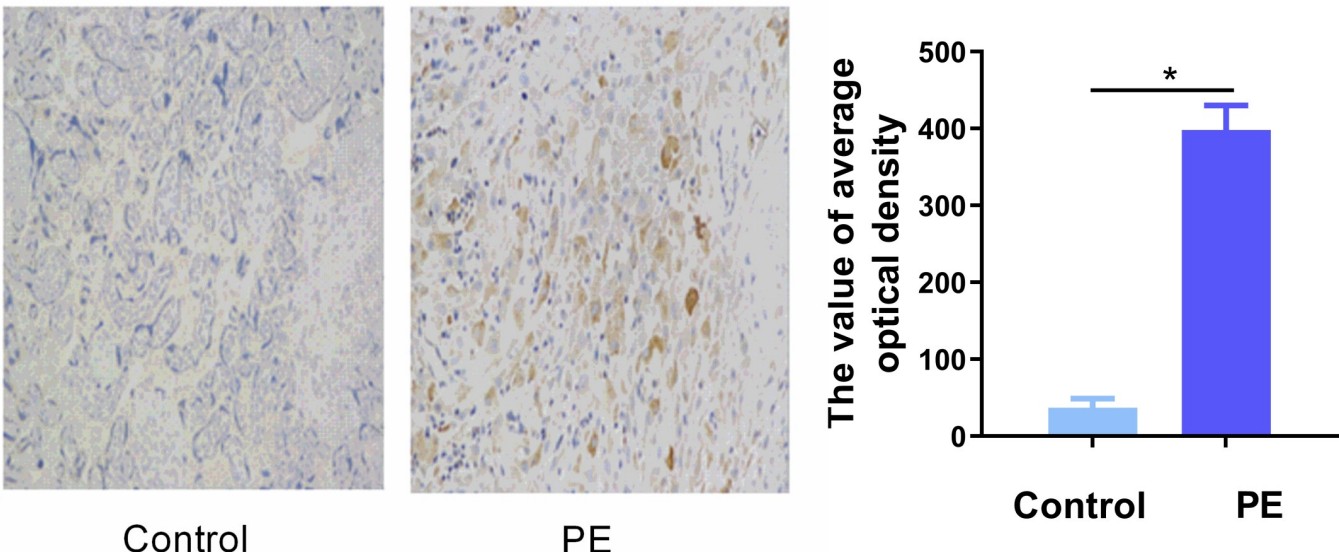

**Fig 3. SerpinA5 immunohistochemistry.** The PE group (n = 10) exhibited strong SerpinA5 staining (average optical density (IOD [mean]) in the placenta compared with that of the control group (n = 10) (DAB, brown, Mann–Whitney U test).

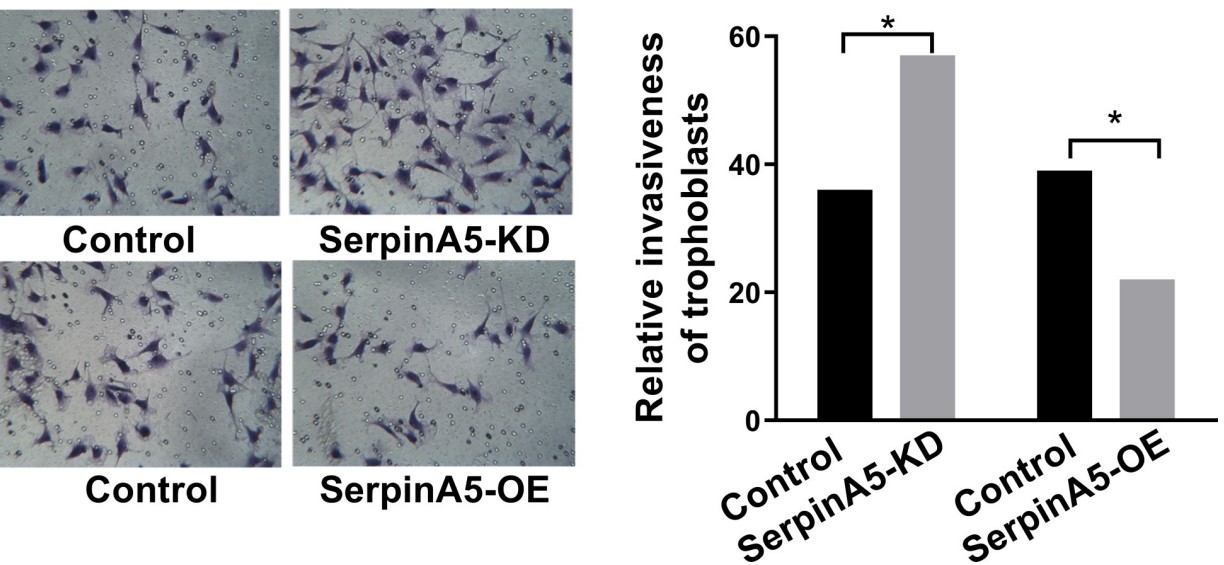

**Fig 4. In vitro invasion analysis.** Over expressed SerpinA5 could inhibit the invasion of trophoblasts; knockdown SERPINA5 could enhance invasion of trophoblasts.

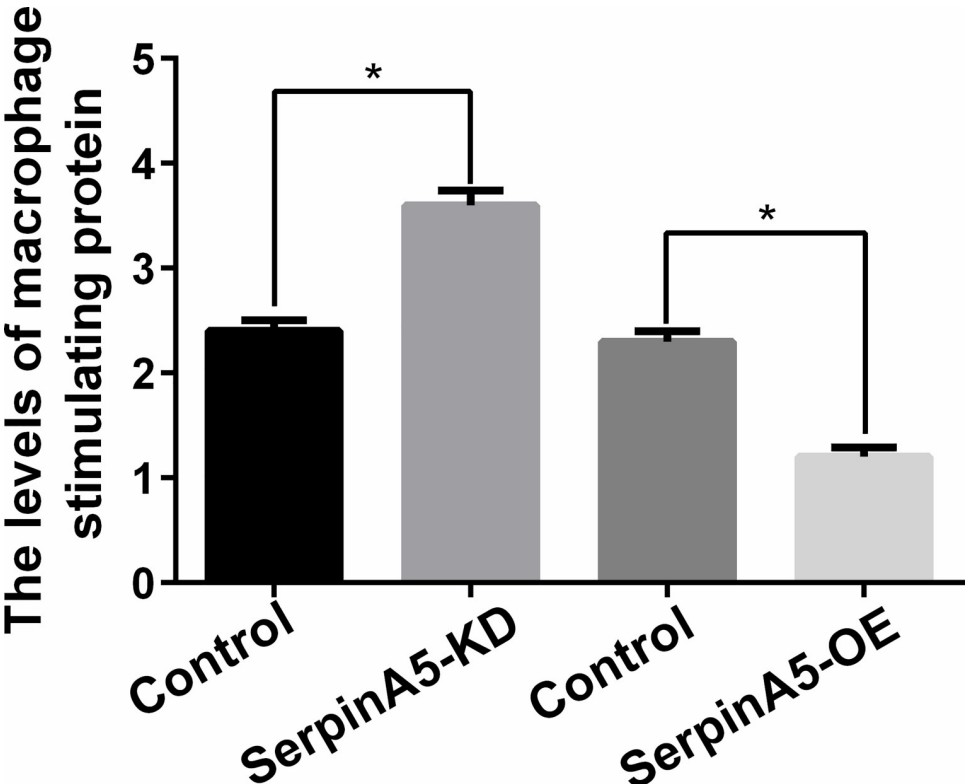

**Fig 5. Simple mechanism analysis.** SerpinA5 may be able to negatively regulate the macrophage stimulating protein (MSP) and MSP could be the target of SerpinA5.

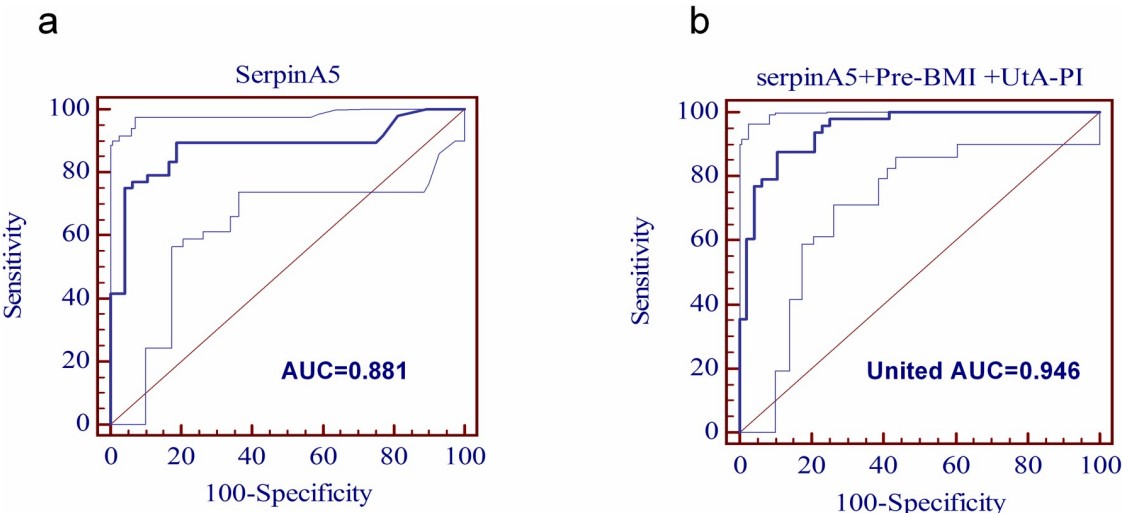

**Fig 6. a. Single Prediction analysis.** Receiver operating characteristic (ROC) curve analysis. As a single predictor SerpinA5, the AUC = 0.881 (95% CI 0.805–0.956)], with 89.58% sensitivity and 81.25% specificity. **b. Combined Prediction analysis.** A combination of plasma SerpinA5, maternal factors and abnormal UtA-PI (above the 95th percentile) enhanced the predictive value for preeclampsia (United AUC was 0.946 (95% CI 0.905–0.988). (Pre-BMI: Pre-pregnancy BMI).

## Discussion

There is a crucial need to identify new biomarkers for the early detection of PE to ensure accurate prediction and better clinical management and to reduce foetal and maternal mortality. Prediction of preeclampsia has always been a hot topic, and there have been many studies on the markers of preeclampsia. In one study, for example, the estimated rate of maternal factors in the prediction of PE was less than 34.0% [8]. Considering this unreliability, many scientists looked for other potential PE prediction factors. Araki et al. screened 23 characteristic peptides in the blood serum of pre-PE patients using quantitative peptidomic analysis [9]. Moreover, Engels et al. reported that the ratio of soluble fms-like tyrosine kinase-1/placental growth factor is a reliable diagnostic tool for PE [10]. Based on the analysis of maternal demographics, medical history and lifestyle, and protein factors, we demonstrated that BMI and MSP were predictors of PE [11]. However, the given prediction significance was relatively low, consistent with previous reports.

Although the pathogenesis of PE is not completely understood, most scholars believe that the pathophysiology of the PE placenta has revealed that the two main pathological characteristics in early pregnancy are disordered remodelling of the uterus spiral arteries and decreased invasion of trophoblast cells [12]. Some substances (such as PIGF, sFLT1, and sENG) from the placenta enter into the maternal circulation, resulting in endothelial cell injury and a series of clinical manifestations of PE [13]. In this study, we also observed differences in 20 plasma proteins between the PE and control groups. Through GO functional analysis, these differentially expressed proteins were involved in various physiological processes, including cell adhesion, apoptosis, inflammatory response, immunity, signal transduction and tissue remodelling. Several proteins play important roles in the extracellular matrix remodelling that is involved in PE development. Based on bioinformatics prediction, we found that a series of serine protease inhibitors were important proteins. In particular, SerpinA5 was the core protein.

Serine protease inhibitors (Serpins, SPI) are a group of proteins with similar structures that include three β-sheets (named A, B and C) and eight or nine α-helices (named hA–hI), yet they are functionally diverse and were first identified as a set of proteins that are able to inhibit

**Table 3. Areas under the operating characteristic curve (AUC) and sensitivity (%) specificity (%) in screening by maternal factors and combinations with uterine artery pulsatility index (UtA-PI) and plasma SerpinA5 in pregnancy.**

| Method of screening | AUC(95% CI) | Sensitivity (%) | Specificity (%) | cut-off value |
|---|---|---|---|---|
| Maternal factors | 0.685(0.579–0.791) | 62.50 | 58.33 | (Pre-pregnancy BMI = 25.0kg/m2 |
| UtA-PI | 0.718 (0.614–0.821) | 56.25 | 87.50 | 2.00 |
| SerpinA5 | 0.881 (0.805–0.956) | 89.58 | 79.22 | 0.070 ng/ml |
| SerpinA5+Pre-pregnancy BMI +UtA-PI | 0.946(0.905–0.988) | 87.50 | 89.58 | |

Mean UtA-PI above the 95th percentile as cut-off value.

proteases. The average protein size of a serpin superfamily member is 350–400 amino acids, but the gene structure varies in terms of the number and size of exons and introns [14]. The majority of the family members have been confirmed to be closely related to preeclampsia [15, 16], such as SerpinA1, A3, E1, and F1. Serpin can perform extracellular roles by inhibiting protease activities in the blood [17]. For example, by inhibiting signalling cascade proteases, extracellular serpins can regulate the proteolytic cascades central to blood clotting (antithrombin), inflammation, immune responses and development [18]. SerpinA5 is also involved in a wide array of physiopathology functions by regulating the activity of other proteases [18, 19].

In this study, many SPIs (such as SerpinA5, SerpinC1, SerpinG1 and SerpinA4) were differentially expressed within PE pregnancies when compared with expression in the control group, but SerpinA5 was significantly different. In previous studies, we confirmed that MSP may play an important role in the pathogenesis of preeclampsia. In this study, we further demonstrated that SerpinA5 levels were greater not only in preeclampsia placental tissue but also in PE gravidas' plasma than levels in the control (P<0.05). We also found that SerpinA5 may interfere with trophoblastic cell invasion by inhibiting MSP (P<0.05). These findings showed that the placenta may release SerpinA5 into the maternal circulation. SerpinA5 has pathological roles in the invasion of trophoblast cells, which is related to the pathological features of preeclampsia. Circulating SerpinA5 may contribute to a novel biomarker for PE.

Despite the prediction efficiency is still insufficient, epidemiological factors and ultrasonographic factors have always been means of predicting preeclampsia [20, 21]. In our results, we also found that BMI and ultrasound arterial pulsatility index have certain predictive value for preeclampsia.

Finally, joint prediction is used. We further found that placenta-derived plasma SerpinA5 together with uterine artery Doppler ultrasound and clinical risk factor can more effectively predict preeclampsia.

The limitations of this paper are that the sample size is not enough lager and the prediction of this disease is not classified into different subtypes.

## Conclusions

In summary, SerpinA5 may be a novel biomarker for preeclampsia and SerpinA5 in conjunction with uterine artery pulsatility index and clinical risk factor may be helpful for the early prediction of preeclampsia.

## Supporting information

**S1 File. Diagnostic criteria and classification definition of preeclampsia.**
(DOCX)

**S2 File. 11–13 weeks specimen matched with maternal age and gestational age of collecting specimens.**
(XLSX)

**S3 File. Screening set data.**
(XLSX)

**S4 File. Developing set data.**
(XLS)

## Acknowledgments

We thank the Director Yang of obstetrics and gynecology and Midwife Zhang for helping to collect placental tissue.

## Author Contributions

**Conceptualization:** Yonggang Zhang, Hongling Yang.

**Data curation:** Yonggang Zhang, Yipeng Zhang, Hongling Yang.

**Formal analysis:** Limin Zhao, Junzhu Shi.

**Funding acquisition:** Yonggang Zhang.

**Writing – original draft:** Yonggang Zhang.

**Writing – review & editing:** Yipeng Zhang, Limin Zhao, Junzhu Shi, Hongling Yang.

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
