## [Decision Letter · Decision Letter 0]

18 Aug 2021

PONE-D-21-13434

Plasma serpinA5 in conjunction with uterine artery pulsatility index and clinical risk factor for the early prediction of preeclampsia

PLOS ONE

Dear Dr. Yonggang Zhang,

Thank you for submitting your manuscript to PLOS ONE. After careful consideration, we feel that it has merit but does not fully meet PLOS ONE’s publication criteria as it currently stands. Therefore, we invite you to submit a revised version of the manuscript that addresses the points raised during the review process.

We look forward to receiving your revised manuscript.

Kind regards,

Maurizio Mandalà, Ph.D.

Academic Editor

PLOS ONE

Additional Editor Comments (if provided):

Reviewers' comments:

Reviewer's Responses to Questions

**Comments to the Author**

1. Is the manuscript technically sound, and do the data support the conclusions?

Reviewer #1: Yes

Reviewer #2: Partly

2. Has the statistical analysis been performed appropriately and rigorously? 

Reviewer #1: Yes

Reviewer #2: Yes

3. Have the authors made all data underlying the findings in their manuscript fully available?

Reviewer #1: No

Reviewer #2: Yes

4. Is the manuscript presented in an intelligible fashion and written in standard English?

Reviewer #1: Yes

Reviewer #2: Yes

5. Review Comments to the Author

Reviewer #1: The manuscript entitled “Plasma serpinA5 in conjunction with uterine artery pulsatility index and clinical risk factor for the early prediction of preeclampsia” evaluated the combined use of plasma protein SerpinA5, uterine artery doppler and clinical risk factors in the prediction of preeclampsia. The study is potentially interesting and well designed, but I have some suggestions before this paper is suitable to for publication as follows bellow.

Abstract: The objective needs to be clearer indicating which plasma protein was evaluated (SerpinA5).

Material and methods and results must inform about the in vitro assay performed with TEV-1 cells.

Keywords: Remove the abbreviation “PE”.

Material and methods:

Line 97: Inform the meaning of “ACOG”. Inform the other characteristics for classification within the preeclampsia group.

Line 115: Inform the proportion of cesarean and natural delivery in each group. Because the authors did not only standardize samples of cesarean deliveries, since natural delivery can alter the expression of some placental proteins.

Line 116: Remove a “)”. It's duplicated.

Lines 145-146: How many fields were evaluated per placenta and which augmentation used.

Line 179: Remove “and”.

Results:

Line 191: Rewrite the sentence. “UtA-PI of preeclampsia was signifcantly diferences than control”.

Subdivide the results into Clinical Characteristics, Proteomics and ELISA, SerpinA5 Immunohistochemistry, In vitro migration analysis, Prediction analysis and further describe each result. The description is superficial and does not adequately describe all results.

Figure captions are uninformative. Need to inform the statistical analysis, N sample, title of the figure. Ex.: What does MSP mean in figure 6? The authors need to graphically demonstrate the immunohistochemical analysis and in vitro migration test.

There are too many figures. Some results can be joined together to reduce the number of figures. (Ex: Figures 7 and 8).

Discussion

Lines 225-227: "Some substances (such as VEGF, FLT, and 226 ENG) from the placenta enter into the maternal circulation, resulting in endothelial cell injury and a series of clinical manifestations of PE." Misinterpretation. Does VEGF Cause Endothelial Cell Injury? sFlt1 and not FLT.

Authors need to better discuss their own results, including clinical data.

Reviewer #2: In the present study, Yonggang Zhang and colleagues examined the expression levels of plasma serpinA5 in order to identify a new biomarker for the early detection of PE. Moreover, the Authors discovered that higher sensitivity and specificity were obtained if plasma serpin A5 was used in combination with maternal factors and abnormal UtA-PI. Finally, Yonggang Zhang and collegues demonstrated that placental serpinA5 immunoistochemistry staining in PE is higher than controls and that SerpinA5 may interfere with trophoblastic cell invasion by inhibiting MSP. They concluded that SerpinA5 may be a novel biomarker for preeclampsia in conjunction with uterine artery pulsatility index and clinical risk factor may be helpful for the early prediction of preeclampsia.

This is an interesting study addressing, however, an extensively investigated issue as biomarkers in early diagnosis of Preeclampsia. Nevertheless, SerpinA5 role and its correlation with UtA-PI and maternal factors in PE placentae could be of interest to Plos One readers.

There are several issues that the Authors must address before publication.

1) The Authors enrolled PE and controls pregnant women however no complete clinical information were reported (e.g. mode of delivery, therapy, fetal sex, placental weight…). Moreover, they must clarify clinical criteria used for PE diagnosis: gestational age at delivery are not significantly different however the fetal weight is significantly decreased in PE. Therefore, did you consider PE with and/or without fetal-placental compromission? Please clarify. Finally, the Authors did not mention the possibility of a fetal-placenta disease in the PE group as the presence of Fetal Growth Restriction. These informations are mandatory to properly analyze and interpret study results.

2) Please add the number of patients included in table 1 and figures 2, 3, 4, 5, 6

3) The Authors performed immunohistochemistry and invasive trial and they demonstrated that SerpinA5 is over-expressed in PE placentae and inhibit MSP leading to a suboptimal trophoblast invasion explaining part of PE pathogenesis. Do you refer to PE with abnormal placental development? Do you think that this mechanism started during the first weeks (first trimester, when you discovered higher plasma SerpinA5?) Why did you chose TEV-1 cells? IHC is a semiquantitative techniques and you cannot conclude that SerpinA5 is over-expressed. You should perform a q quantitative techniques such as Real Time PCR and ELISA assay. Please, clarify

4) Please, better specify why SerpinA5 is better than sFlt-1/PlGF (e.g. higher sensitivity and specificity? Earlier diagnosis?)

6. PLOS authors have the option to publish the peer review history of their article (what does this mean?). If published, this will include your full peer review and any attached files.

Reviewer #1: No

Reviewer #2: No

---

## [Author Response · Author response to Decision Letter 0]

26 Aug 2021

Replies to Reviewers:

Reviewer's Responses to Questions

1. Is the manuscript technically sound, and do the data support the conclusions?

Reviewer #1: Yes

Reviewer #2: Partly

 Response: We would like to express our sincere gratitude to the reviewers for your constructive and positive comments.

 2. Has the statistical analysis been performed appropriately and rigorously?

Reviewer #1: Yes

Reviewer #2: Yes

 Response: We would like to express our sincere gratitude to the reviewers for your constructive and positive comments.

3. Have the authors made all data underlying the findings in their manuscript fully available?

Reviewer #1: No

Reviewer #2: Yes

 Response: We would like to express our sincere gratitude to the reviewers for your constructive and positive comments. We have added the corresponding other data in the supporting information.

4. Is the manuscript presented in an intelligible fashion and written in standard English?

Reviewer #1: Yes

Reviewer #2: Yes

 Response: We would like to express our sincere gratitude to the reviewers for your constructive and positive comments.

Replies to Reviewer 1

Reviewer #1: The manuscript entitled “Plasma serpinA5 in conjunction with uterine artery pulsatility index and clinical risk factor for the early prediction of preeclampsia” evaluated the combined use of plasma protein SerpinA5, uterine artery doppler and clinical risk factors in the prediction of preeclampsia. The study is potentially interesting and well designed, but I have some suggestions before this paper is suitable to for publication as follows bellow.

Response: 

Thank you very much for your positive comment on the study and insightful suggestion. We have revised the manuscript, point-by-point responses to the comments are listed below. I hope these replies would satisfy you.

1). Abstract: The objective needs to be clearer indicating which plasma protein was evaluated (SerpinA5). Material and methods and results must inform about the in vitro assay performed with TEV-1 cells.

Response: 

Thank you for your helpful suggestion. We have made the relevant corrections as required (Page 2 Line 31 and Page 2 Lines 43-44 and Page 2 Lines 47-48).

2). Keywords: Remove the abbreviation “PE”.

Response: Thank you very much for your serious and meticulous work！

We apologize for our negligence. This correction has been made (Page 3 Line 56).

3). Material and methods:

Line 97: Inform the meaning of “ACOG”. Inform the other characteristics for classification within the preeclampsia group.

Line 115: Inform the proportion of cesarean and natural delivery in each group. Because the authors did not only standardize samples of cesarean deliveries, since natural delivery can alter the expression of some placental proteins.

Line 116: Remove a “)”. It's duplicated.

Lines 145-146: How many fields were evaluated per placenta and which augmentation used.

Line 179: Remove “and”.

Response: Thank you for your valuable suggestion.

The modifications have been made into the revised manuscript. 

ACOG is American College of Obstetricians and Gynecologists (Page 4 Lines 87).

The other characteristics for classification within the preeclampsia group was showed in Supporting information S1 File from Ref 1 as below [1]. 

The proportion of cesarean and natural delivery in each group was showed in Table 1.

Per section and three sections each specimen and ten randomly selected fields were evaluated and no augmentation was used in immunohistochemistry (Page 6 Lines 137-138). and other repeated or incorrect wording was deleted (Page 5 Line 108 and Page 8 Line 172).

We hope that the revisions satisfy the above-mentioned requirements.

[1]. Hypertension in pregnancy. Report of the American College of Obstetricians and Gynecologists’ Task Force on Hypertension in Pregnancy. Obstet Gynecol. 2013 Nov;122(5):1122-1131. 

4). Results:

Line 191: Rewrite the sentence. “UtA-PI of preeclampsia was signifcantly diferences than control”.

Subdivide the results into Clinical Characteristics, Proteomics and ELISA, SerpinA5 Immunohistochemistry, In vitro migration analysis, Prediction analysis and further describe each result. The description is superficial and does not adequately describe all results.

Figure captions are uninformative. Need to inform the statistical analysis, N sample, title of the figure. Ex.: What does MSP mean in figure 6? The authors need to graphically demonstrate the immunohistochemical analysis and in vitro migration test.

There are too many figures. Some results can be joined together to reduce the number of figures. (Ex: Figures 7 and 8).

Response: Thank you for your insightful and good suggestion.

The modifications have been made into the revised manuscript. 

The average UtA-PI of preeclampsia was significantly higher than that control during the first trimester (Rewrite the sentence in Page 9 Lines 186-187).

The results have been subdivided into Clinical Characteristics, Proteomics and ELISA, SerpinA5 Immunohistochemistry, In vitro migration analysis, Prediction analysis (Page 8 Line 183 and Page 9 Lines 188, 195, 199, 206) and further described (Page 9 Lines 196,197, 200-205 and Page 10 Lines 207-209). 

Figure captions and Figure have been revised as required. 

MSP is level of macrophage stimulating protein (Page 19 Line 420). 

Statistical analysis and N sample have been added into Figure legends 2-3(Page 19 Line 406-415). Titles of the figure have been added into Figure legends 1-6 (Page 19 Line 406-425).

The immunohistochemical analysis and in vitro migration test have been graphically presented in Fig 3, 4.

We hope that all the revisions satisfy the above-mentioned requirements.

5). Discussion

Lines 225-227: "Some substances (such as VEGF, FLT, and 226 ENG) from the placenta enter into the maternal circulation, resulting in endothelial cell injury and a series of clinical manifestations of PE." Misinterpretation. Does VEGF Cause Endothelial Cell Injury? sFlt1 and not FLT.

Authors need to better discuss their own results, including clinical data.

Response: 

Thank you very much for your good suggestion. The modifications have been made to the revised manuscript (Page 11 Line 234). We apologize for our inaccurate language expression, actually, we mean it is sFlt1 and not FLT.

The discussion section has been modified (Page 12 Lines 268-271). We hope that the revisions satisfy your requirements.

Replies to Reviewer 2

Reviewer #2: In the present study, Yonggang Zhang and colleagues ……. They concluded that SerpinA5 may be a novel biomarker for preeclampsia in conjunction with uterine artery pulsatility index and clinical risk factor may be helpful for the early prediction of preeclampsia.

This is an interesting study addressing, however, an extensively investigated issue as biomarkers in early diagnosis of Preeclampsia. Nevertheless, SerpinA5 role and its correlation with UtA-PI and maternal factors in PE placentae could be of interest to Plos One readers.

There are several issues that the Authors must address before publication.

Response: Thank you very much for your positive comment on the study and insightful suggestion for further investigation. We have revised the manuscript, point-by-point responses to the comments are listed below. I hope these replies would satisfy you.

1) The Authors enrolled PE and controls pregnant women however no complete clinical information were reported (e.g. mode of delivery, therapy, fetal sex, placental weight…). Moreover, they must clarify clinical criteria used for PE diagnosis: gestational age at delivery are not significantly different however the fetal weight is significantly decreased in PE. Therefore, did you consider PE with and/or without fetal-placental compromission? Please clarify. Finally, the Authors did not mention the possibility of a fetal-placenta disease in the PE group as the presence of Fetal Growth Restriction. These informations are mandatory to properly analyze and interpret study results.

Response: Thank you for your insightful and valuable suggestion. I can't agree with you more. The amount and proportion of mode of delivery has been added Table 1. There was no difference in fetal sex between the two groups. Because it's not our focus, we're very apologetic that we didn't collect the data of the therapy, and placental weight. 

Gestational age of collecting specimen is very important for the expression of plasma and placental factors. These factors are not only affected by pathological changes, but also may change greatly with gestational weeks. Therefore, the gestational weeks matching between control group and PE group is very important. As reported in a literature [2], unexplained preterm birth (PTL) is a better control group for PE group with small gestational weeks. So, in the control, including normal pregnancy group and PTL. Thus , the gestational age at delivery is not significantly different. And no Fetal Growth Restriction in PE group (Table1).

[2]. Zhou Y, Bianco K, Huang L, Nien JK, McMaster M, Romero R, Fisher SJ. Comparative analysis of maternal-fetal interface in preeclampsia and preterm labor. Cell Tissue Res. 2007 Sep;329(3):559-69. 

2) Please add the number of patients included in table 1 and figures 2, 3, 4, 5, 6

Response: the number of patients has been added included in new Table 1 and Figure legends 2, 3(Page 19 Lines 406-415); and the assays in Figures 3, 4, 5 was repeated three times and described in Page 6 Lines 137-138 and Page 7 Line 159.

3) The Authors performed immunohistochemistry and invasive trial and they demonstrated that SerpinA5 is over-expressed in PE placentae and inhibit MSP leading to a suboptimal trophoblast invasion explaining part of PE pathogenesis. Do you refer to PE with abnormal placental development? Do you think that this mechanism started during the first weeks (first trimester, when you discovered higher plasma SerpinA5?) Why did you chose TEV-1 cells? IHC is a semiquantitative techniques and you cannot conclude that SerpinA5 is over-expressed. You should perform a q quantitative techniques such as Real Time PCR and ELISA assay. Please, clarify

Response: 

Thank you for raising this issue. 

Placental theory holds that preeclampsia is mainly caused by poor placental formation. Preeclampsia occurs after 20 weeks. However, there are differences in some plasma factors of placenta before 20 weeks of pregnancy or even during the first pregnancy. SerpinA5 is over-expressed in PE placentae after delivery (>20 weeks). The onset and development of preeclampsia is very complex, here we are just making a very simple preliminary exploration of SerpinA5 in PE. The main purpose of this study is to do prediction analysis for PE, not mechanism research for PE.

Hui Chen Feng et al [3]. transfected the primary trophoblast isolated from the placenta of early pregnancy with the viral vector of HPV E16 / E17 gene to immortalize the cells. The phenotypic identification methods such as immunohistochemistry, RT-PCR and western blotting showed that TEV-1 (human extravillous trophoblast cell line) cell line maintained all the markers of extravillous trophoblasts. Therefore, it can be said that the immortalized cell line. TEV-1 cells maintain the characteristics of extravillous trophoblasts and can express invasive proteins. TEV-1 cell line can be used as a cell model to study the physiological and functional characteristics of human extravillous trophoblasts [3].

Thank you for your positive comment on the study and insightful suggestion for further investigation. In the next mechanism research, we are planning a further study to demonstrate the relationship between placental expression of SerpinA5 and preeclampsia by western blot analysis, and we prepare to do more in vivo and in vitro mechanism research. In IHC analysis in the present study, 10 microscopic fields were randomly selected in each section. Image Pro-Plus 6.0 software, which is useful for localization and semi-quantitative analysis, was used to quantify the integrated optical densities (IODs) of the SerpinA5-positive reactions.

[3].Feng HC, Choy MY, Deng W, Wong HL, Lau WM, Cheung AN, Ngan HY, Tsao SW. Establishment and characterization of a human first-trimester extravillous trophoblast cell line (TEV-1). J Soc Gynecol Investig. 2005 May;12(4):e21-32.

4) Please, better specify why SerpinA5 is better than sFlt-1/PlGF (e.g. higher sensitivity and specificity? Earlier diagnosis?)

Response: 

Thank you for your constructive suggestion

The results are need to be validated in larger numbers. but the prediction effectiveness of sigle plasma protein SerpinA5 is insufficient, and plasma protein alone(sFlt-1/PlGF) is not enough (N Engl J Med. 2016; 374(18):1785-1786.) As stated in ACOG 2013, the combination of multiple prediction means is required for PE prediction. This study combined with a new plasma protein SerpinA5, ultrasonic Doppler and clinical risk factors to predict preeclampsia, which provides a new way for the prediction of preeclampsia.

---

## [Editor Report · Decision Letter 1]

30 Sep 2021

Plasma serpinA5 in conjunction with uterine artery pulsatility index and clinical risk factor for the early prediction of preeclampsia

PONE-D-21-13434R1

Dear Dr. Yonggang Zhang,

We’re pleased to inform you that your manuscript has been judged scientifically suitable for publication and will be formally accepted for publication once it meets all outstanding technical requirements.

Kind regards,

Maurizio Mandalà, Ph.D.

Academic Editor

PLOS ONE
---

## [Editor Report · Acceptance letter]

6 Oct 2021

PONE-D-21-13434R1 

Plasma serpinA5 in conjunction with uterine artery pulsatility index and clinical risk factor for the early prediction of preeclampsia 

Dear Dr. Zhang:

I'm pleased to inform you that your manuscript has been deemed suitable for publication in PLOS ONE. Congratulations! Your manuscript is now with our production department. 

Kind regards, 

on behalf of

Dr. Maurizio Mandalà 

Academic Editor

PLOS ONE